# The Seaweed Diet in Prevention and Treatment of the Neurodegenerative Diseases

**DOI:** 10.3390/md19030128

**Published:** 2021-02-26

**Authors:** Leonel Pereira, Ana Valado

**Affiliations:** 1Department of Life Sciences, University of Coimbra, 3000-456 Coimbra, Portugal; 2MARE-Marine and Environmental Sciences Centre, Faculty of Science and Technology, University of Coimbra, 3000-548 Coimbra, Portugal; 3Polytechnic Institute of Coimbra, ESTESC-Coimbra Health School, R. 5 de Outubro, S. Martinho do Bispo, Ap. 7006, 3046-854 Coimbra, Portugal; valado@estescoimbra.pt

**Keywords:** edible macroalgae, public health, nutraceuticals, functional foods, neuroprotective agents

## Abstract

Edible marine algae are rich in bioactive compounds and are, therefore, a source of bioavailable proteins, long chain polysaccharides that behave as low-calorie soluble fibers, metabolically necessary minerals, vitamins, polyunsaturated fatty acids, and antioxidants. Marine algae were used primarily as gelling agents and thickeners (phycocolloids) in food and pharmaceutical industries in the last century, but recent research has revealed their potential as a source of useful compounds for the pharmaceutical, medical, and cosmetic industries. The green, red, and brown algae have been shown to have useful therapeutic properties in the prevention and treatment of neurodegenerative diseases: Parkinson, Alzheimer’s, and Multiple Sclerosis, and other chronic diseases. In this review are listed and described the main components of a suitable diet for patients with these diseases. In addition, compounds derived from macroalgae and their neurophysiological activities are described.

## 1. Introduction

Neurodegenerative diseases are pathologies characterized by the irreversible destruction of certain neurons, which leads to the progressive and disabling loss of certain functions of the nervous system. Some of them are now considered the biggest causes of dementia in the world [1]. It is estimated that 9.9 million European citizens and 35.6 million worldwide suffer from some form of dementia. These figures are expected to double in 2030, and triple in 2050. People living with dementia have little access to adequate health care, even in most high-income countries, where only about 50% of people living with dementia are properly diagnosed and followed up. In low- and middle-income countries, less than 10% of cases are diagnosed. As the population ages, due to the increase in life expectancy, the number of people with dementia is increasing. Although the neurodegenerative diseases of Alzheimer’s (AD) and Parkinson’s (PD) are forms of dementia, there are other syndromes that have similar symptoms―such as depression, hallucinations, and memory loss―syndromes that include dementia of Lewy bodies, vascular dementia, frontotemporal dementia, etc. [2].

For this reason, much attention has been paid by scientists to safe and efficient neuroprotective agents. Several categories of natural and synthetic neuroprotective agents have been described. However, it is believed that synthetic neuroprotective agents may have side effects, such as tiredness, drowsiness, numbness in the upper and lower limbs, balance difficulties, nervousness, or anxiety, etc. [3]. Thus, currently, researchers are interested in evaluating natural bioactive compounds that can act as neuroprotective agents. Adaptogens comprise a category of medicinal and nutritional products based on plants that promote adaptability, resilience, and survival of living organisms under stress. Common adaptogenic plants used in various traditional medical systems (TMS) and conventional medicine provide a modern justification for their use in the treatment of stress-induced and age-related diseases [4]. In this sense, seaweed can be a potential source of neuroprotective agents [5].

The development of neuroprotective agents from seaweed still faces several challenges. The justification for the treatment of the neuroprotective effects of marine algae on the central nervous system (CNS) has been based on observations and experiments established in vitro or only on animal models. So far, few of the neuroprotective effects of seaweed have been studied directly in humans [6]. Thus, more clinical studies and other large-scale controlled studies are needed [7].

Another major challenge in the development of studies on marine algae to obtain neuroprotective agents is that many of the drugs currently available are unable to provide effective neuroprotection. Possible reasons for this failure include the inappropriate use of specific medication for a particular disease, or the disease’s progression stage was too advanced, or suboptimal doses were used. Thus, future studies need to focus on the synergistic benefits of consuming different species of seaweed (or their extracts), recommended doses, ingestion times, and preparation methods for bioactive seaweed compounds to maximize the desired protective effect in preventing neurodegenerative diseases [8,9].

Currently, several lines of study attempt to provide information on the biological activities and neuroprotective effects of seaweed, including antioxidants, anti-neuroinflammatory agents, cholinesterase inhibitory activity, and inhibition of neuronal death [5]. Recent studies have shown that microglia activation and the resulting production of pro-inflammatory and neurotoxic factors are sufficient to induce neurodegeneration in animal models. In addition, microglia activation and excessive amounts of proinflammatory mediator release by microglia were observed during the pathogenesis of AD, PD, MS, dementia complex, as well as neuronal post-death in strokes and traumatic brain injuries (Figure 1). Therefore, a mechanism to regulate the release of the inflammatory response by microglia may have important therapeutic potential for the treatment of neurodegenerative diseases. Several published works point out that seaweed constitutes a relevant source of neuroprotective agents, with particular interest for preventive therapeutics (Table 1) [5].

## 2. Alzheimer’s and Parkinson’s Diseases

Typically, afflicting adults in mid-life, neurodegenerative diseases are characterized by motor or cognitive changes that get progressively worse with age, and that usually reduce life expectancy. Human neurodegenerative disease results from the influence of several environmental and genetic causes [10]. Among the series of identified disease conditions that relate to extensive loss of function and quantity of neurons, Parkinson’s and Alzheimer’s are well-defined neurodegenerative diseases. However, the etiology of those diseases has not yet been identified clearly, even though some of them were reported centuries before [11].

Currently, with increasing life expectancy and demographic changes in the population, neurodegenerative diseases, such as AD and PD, are becoming frighteningly common [12,13]. AD is the most common form of dementia, accounting for about 50 to 70% of all cases.

AD is a type of dementia that causes a global, progressive, and irreversible deterioration of various cognitive functions (memory, attention, concentration, language, thinking, among others). This deterioration results in changes in behavior, personality, and functional capacity of the person, making it difficult to perform their daily activities. The name of this disease is due to Alois Alzheimer, a German doctor who in 1907 first described the disease [14].

In AD, dysregulation of the level of beta-amyloid (Aβ) leads to the appearance of senile plaques that contain depositions of Aβ. Aβ is a complex biological molecule that interacts with many types of receptors and/or forms insoluble assemblies and, eventually, its non-physiological depositions alternate with normal neuronal conditions [15].

In neuropathological terms, AD is characterized by neuronal death in certain parts of the brain, with some causes yet to be determined. The appearance of fibrillary braids and senile plaques make communication between nerve cells impossible, which causes changes in the overall functioning of the person. In the early stages, the symptoms of AD can be very subtle. However, they often begin with lapses of memory and difficulty in finding the right words for everyday objects. These symptoms worsen as brain cells die and communication between them is altered [16].

Decreased levels of the neuro-mediators acetylcholine (ACh) and butyrylcholine (BCh) have been observed in the brains of patients with AD. For this reason, inhibition of Acetylcholinesterase enzyme (AChE) and Butyrylcholinesterase enzyme (BuChE), responsible for the hydrolysis of ACh and BCh, has become a treatment option for AD [17].

Involvement of reactive oxygen species (ROS), such as hydrogen peroxide (H_2_O_2_), has been suggested in diseases such as AD, schizophrenia, amyotrophic lateral sclerosis, PD, and other degenerative diseases of the basal ganglia, systemic atrophy, and multiple and progressive supranuclear degeneration are theorized by having the activity of free radicals as mediators. The neurotoxicity of H_2_O_2_ is mainly exerted by the formation of the highly reactive hydroxyl (OH•) radical, although the depletion of glutathione (GSH) levels and the secondary rupture of calcium homeostasis may also contribute to the toxic effects of H_2_O_2_ [18,19,20]. Fallarero et al. [21] showed that *Halimeda incrassata* (Chlorophyta) is an effective ROS scavenger in mouse hypothalamic (GT1–7) cells.

PD is a degenerative and slowly progressive disease of specific areas of the central nervous system (brain and spinal cord). It is characterized by tremors when the muscles are at rest (resting tremor), increased muscle tone (stiffness), slow movement of the muscles, and difficulty maintaining balance (postural instability). In many people, thinking becomes compromised. In PD, nerve cells and part of the basal ganglia (called black substance) degenerate. Basal ganglia are structures related to the movement, although they do not send connections directly to the spinal cord or cranial nerves. Functionally, the following structures are part of the basal ganglia: The caudate nucleus, the putamen, the pale globe, the subthalamic nuclei, and the substantia nigra. Like all nerve cells, those of the basal ganglia release chemical messengers, the neurotransmitters, which stimulate the next nerve cell (neuron) and allow it to send a nerve impulse. Dopamine is the major neurotransmitter used in the basal ganglia. Its general effect is to intensify the nerve impulses so that they reach the muscles. Thus, the fundamental functions of the base ganglia are related to the cognitive part of the movement, such as planning and performing complex motor acts. Its dysfunction determines changes in the reciprocal control of muscles, causing stiffness, tremors, and akinesia [5,22].

## 3. Seaweeds and Their Neurophysiological Activities

Macroalgae-derived compounds with neuroprotective activity may provide some important nutrients for the prevention and treatment of neurodegenerative diseases such as AD, PD, and other neurodegenerative diseases. Much of these bioactive compounds are derived from Phaeophyceae, brown algae (57.6%), followed by Rhodophyta, red algae (28.3%) and Chlorophyta, green algae (14.1%) [23] (Table 1).

Among the various components, carbohydrates are the most abundant constituents of seaweed. In addition, polysaccharides are generally the main component of red, green, and brown algae [24], and monosaccharides and oligosaccharides are also present. The reserve polysaccharides are laminarin in brown algae, floridean starch (more branched than amylopectin) in red algae, and starch in green algae. The algal cell walls are characterized by the presence of unusual polysaccharides that can be sulfated, acetylated, etc. Thus, seaweed carbohydrates are promising compounds in several fields, such as food, pharmaceutical, and biomedical [24,25,26]. The therapeutic applications of these notable polysaccharides are, among others, antiviral, antibacterial, and antitumor activity, antioxidant, antidiabetic, antilipidemic properties, anti-inflammatory, and immunomodulatory characteristics [24,27]. Studies related to the absorption of bioactive compounds extracted from algae, namely polysaccharides, provide vital information on the most appropriate administration routes. If a drug formulation has a high absorption rate, it can be used for the delayed/controlled release of an active ingredient. Understanding the pharmacokinetics of polysaccharides derived from seaweeds (alginates, laminarins, fucoidans, etc.), may lead to their extensive use not only as drugs, but also to improve the bioavailability of certain poorly soluble compounds in pharmaceutical formulations (Shikov et al., 2020) [28]. Laminarin, a polysaccharide composed of (1,3)-β-d-glucan with some β (1,6) branching, particularly abundant in species of the genus *Laminaria* (Figure 2a) [24], has been shown to have antibacterial and chemo-preventive activities, along with pre-biotic, important in the modulation of the intestinal microbiota, which in turn can regulate neuro-inflammation [25].

PD is generally characterized, as we have seen previously, by the loss of dopaminergic neurons, and the presence of 1-methyl-4-phenyl-1,2,3,6-tetrahydropyridine (MPTP) can induce PD [29]. The administration of this substance may result in motor dysfunction, such as occurs in PD, which makes it a suitable experimental model for this disease [30].

A compound (fucoidan) has been found to attenuate the neurotoxicity of MPTP activity. This sulfated polysaccharide, derived from *Saccharina japonica* (Phaeophyceae), has been demonstrated in mice models to be effective at a dosage of 25 mg kg^−1^ in protecting the cells from MPTP-induced neurotoxicity by reducing the behavioral deficits and cell death and increasing the level of dopamine [31]. Tau is a microtubule-associated protein (MAP) found in axons, and this protein is responsible for regulating the stability of microtubules [32,33,34]. Hyper-phosphorylation of tau results in its dissociation from microtubules and aggregation in the form of neurofibrillary tangles [35]. Hyper-phosphorylated tau protein is a major component in neurofibrillary tangles, which is a hallmark of AD, and dysregulation of kinases and phosphatases has been found to increase tau hyper-phosphorylation levels [36,37]. Only three compounds (Spiralisone A, B, and Chromone 6) from seaweed have kinase inhibitory activity, and these compounds have been isolated from brown algae (Phaeophyceae). Besides, these compounds were isolated from a single species of brown algae, the *Zonaria spiralis* harvested in Australia, and all of them are phloroglucinols. The most active compound is spiralisone B, inhibiting the kinases-cyclin-dependent kinase 5 (CDK5/p25), casein kinase 1 (CK1δ), and glycogen synthase kinase 3β (GSK3β), with IC_50_ values of 3, 5, and 5.4 μM, respectively [38].

Polysaccharide extracts from seaweed have very significant neuroprotective and repairing activities. These polysaccharides could be the next great advance in the treatment of neurodegenerative diseases. Fucoidan, ulvan, and their derivatives are potential agents to treat Alzheimer’s disease, according to a recent review made by Bauer et al. [39].

A study carried out by Jhamandas et al. [40] successfully showed that fucoidan isolated from *Fucus vesiculosus* (Figure 2b) (Phaeophyceae), could protect cholinergic neuronal death in rats induced by the beta amyloid protein (Aβ1). Fucoidan pretreatment blocked the activation of caspase-9 and caspase-3. Caspase-9 and caspase-3 have been suggested to mediate the terminal stages of neuronal apoptosis [41]. Therefore, fucoidan’s ability to block the activation of caspase-9 and caspase-3 suggests that the inhibition of neuronal death, promoted by fucoidan, occurs mainly through apoptotic inhibition. In neurodegenerative diseases, apoptosis may be pathogenic, and targeting this process may mitigate neurodegenerative diseases [42].

Fucoidans extracted from the brown algae *Laminaria hyperborea* and *Saccharina latissima* have recently been suggested as potential therapeutics in age-related macular degeneration (AMD), and in other pathological pathways that include lipid dysregulation, inflammation, oxidative stress, and pro-angiogenic signaling [43]. However, knowing the pharmacokinetic profile of these bioactive molecules, including their molecular weight, is essential to implement drug development processes [44].

Eight compounds have been found from macroalgae (Phaeophyceae and Chlorophyta) with neuroprotective activity against beta amyloid protein (Aβ), and five compounds of them are extracted from the green alga *Caulerpa racemosa* (Figure 2c) [22]. 

In the study made by Najam et al. [45], the administration of methanolic extract of *Hypnea musciformis* (Figure 2d) (Rhodophyta) significantly increased the level of dopamine on rats and mice. The possible effect of *H. musciformis* on dopamine and other biogenic amines in the brain indicates that *H. musciformis* probably has a psychotropic and anxiolytic profile. Increasing the level of dopamine may also be beneficial in view of the etiology of PD. In this study, the serotonin level was decreased after the administration of *H. musciformis*. The regular use of algae as a diet alleviates the symptoms of anxiety because the known anxiolytics also manifest their effect by reducing the concentration of serotonin [45].

Mohibbullah et al. [46] collected 23 edible seaweeds from Korean and Indonesian coasts to screen for marine seaweeds with potent neuroprotective activity. Hippocampal neurons of rats (DIV 9) were cultured in the presence of ethanol extracts and the cultures were treated with three different concentrations of seaweed extract: 5, 15, and 30 µg mL^−1^. About 1/3 of the tested seaweeds exhibited neuroprotective activity. Cell viability and cell cytotoxicity testing revealed that the ethanol *Gracilariopsis chorda* (Rhodophyta) extract afforded the most neuroprotection at a concentration of 15 µg mL^−1^, at which *G. chorda* significantly increased cell viability to 3.2–119.0%, and decreased cell death to 10.3–80.5%. *Undaria pinnatifida* (brown alga) had almost the same level of neuroprotection as *G. chorda*, and others like *Sargassum fulvellum, Sargassum nigrifolium* (brown algae), *Neopyropia yezoensis* (as *Porphyra yezoensis*), *Gracilaria coronopifolia*, *Agarophyton tenuistipitatum* (formerly *Gracilaria tenuistipitata*) (red algae), *Ecklonia bicyclis* (as *Eisenia bicyclis*) (brown alga), and *Grateloupia cornea* (formerly *Carpopeltis cornea*) (red alga) also exhibited moderate neuroprotective effects [46]. 

Liu et al. [47] demonstrated that dietary supplementation of the worms with an extract from the cultivated red seaweed *Chondrus crispus* (Figure 2e) (Rhodophyta) decreased the accumulation of α-synuclein and protected the worms from the neuronal toxin-induced 6-hydroxy-dopamine (6-OHDA model), and the brought dopaminergic neurodegeneration. These effects were associated with a corrected slowness of movement. The authors also showed that increased tolerance to oxidative stress and a positive regulation of the stress response genes, sod-3 and skn-1, may have served as a molecular mechanism for the protection mediated by *C. crispus* extract against pathology’s of PD. In all, in addition to its potential as a functional food, the tested red algae, *C. crispus*, may find promising pharmaceutical applications for the creation of novel anti-neurodegenerative drugs [47]. 

According to Tirtawijaya et al. [48], ethanolic extract of *Kappaphycus alvarezii* (Figure 2f) (Rhodophyta) significantly increased numbers of axodendritic intersections, branching points and branching tips, in cultures of fetal rat hippocampal neurons. Thus, *K. alvarezii* may be useful as a diet supplement or pharmaceuticals for people who are prone to neurological disorders.

In the works of Silva and collaborators [49], the neuroprotective effects of the *Bifurcaria bifurcata* (Figure 2g) (Phaeophyceae) extracts were evaluated in a neurotoxic model induced by 6-hydroxydopamine (6-OHDA) in a human neuroblastoma cell line (SH-SY5Y), while the mechanisms associated to neuroprotection were investigated by the determination of mitochondrial membrane potential, H_2_O_2_ production, Caspase-3 activity, and by observation of Deoxyribonucleic Acid (DNA) fragmentation. The extracted diterpenes eleganolone and eleganonal exhibited antioxidant potential, being interesting candidates for further neuroprotective studies [49].

A food supplement approved by the Food and Drug Administration (FDA), named Aquamin, is a natural multi-mineral derived from the marine edible red macroalgae *Lithothamnion corallioides* (Figure 2h). Aquamin was evaluated for its anti-neuroinflammatory potential, and in cortical glial-enriched cells was able to suppress the release of lipopolysaccharides (LPS)-induced tumor necrosis factor (TNF)-α and interleukin (IL)-1β. Several authors suggested that anti-inflammatory and antioxidative agents could prevent the deposition of Aβ and the subsequent brain damage [24,50].

According to Olasehinde et al. [51], aqueous ethanol extracts rich in phlorotannins, phenolic acids and flavonoids from *Gracilaria gracilis* (Rhodophyta) (Figure 2n) and *Ulva lactuca* (Chlorophyta) (Figure 2o) exhibited AChE and BChE inhibitory activities. In addition, the sulfated polysaccharides obtained from *Ulva rigida*, as well as the algae species mentioned above, also showed potent inhibitory effects on BChE and AChE in vitro.

**Table 1 marinedrugs-19-00128-t001:** Bioactive properties of some compounds extracted from seaweeds.

Species	Extraction Methods	Compounds of Interest and Fractions	Activity		References
**Phaeophyceae (brown seaweeds)**					
*Agarum clathratum* subsp. *yakishiriense*	Ethanol extract at 60 °C for 2 h	The ethanol extract was suspended in distilled water and subjected to a series of partitioning with *n*-hexane, dichloromethane, ethyl acetate and n-butanol; the mass of crude extract (95% EtOH) and fractions was 840.46 mg	In vivo (animal models) neuronal protection from ischemic injury		[52]
*Alaria esculenta*	Extracted with methanol/water (1:1) at 50 °C with stirring for 2 h at room temperature	Fractions	The fraction below 5 kDa decreased the melting point of α-synuclein, whereas the fraction above 10 kDa raised the melting point. Both of these fractions were found to inhibit the formation of amyloid aggregates by α-synuclein, in vitro		[53]
*Cystoseira humilis*	Methanolic extract	Fraction	In vitro AChE inhibitory capacity: 50% 10 mg mL^−1^		[54]
*Gongolaria nodicaulis* (as *C. nodicaulis*)	Methanolic extract	Fraction	In vitro AChE inhibitory capacity: 64.4% In vitro BuChE inhibitory capacity: 110% 10 mg mL^−1^		[54]
*Ericaria selaginoides* (as *Cystoseira tamariscifolia*)	Methanolic extract	Fraction	In vitro AChE inhibitory capacity: 85% In vitro BuChE inhibitory capacity: 86% 10 mg mL^−1^		[54]
*Gongolaria usneoides* (as *Cystoseira usneoides*)	Methanolic extract	Fraction	In vitro AChE inhibitory capacity: 47% 10 mg mL^-1^		[54]
*Dictyopteris undulata*	Zonarol was prepared as a 10 mM stock solution in dimethyl sulfoxide (DMSO)	Sesquiterpene: Zonarol	Activates the Nrf2/ARE pathway, induces phase-2 enzymes, and protects neuronal cells from oxidative stress, in vitro		[55]
*Dictyota humifusa*	Methanolic extract	Extract	Inhibiting AChE IC_50_ = 4.8 mg mL^−1^, in vitro		[56]
*Ecklonia bicyclis* and *E. bicyclis* (as *Eisenia bicyclis*)	MeOH extract	Phlorotannins	Suppression of BACE-1 enzyme activity IC_50_ = 5.35 μM, in vitro		[57]
	Ethyl acetate extraction	Phlorotannins	Decreased Aβ-induced cell death IC_50_ = 800 µM, in vivo		[58]
	Ethanolic extract	Phlorotannins	Protection from retinal neuronal death, in vivo		[59]
	Ethanolic extract	Phlorotannins	In vitro inhibitory properties against AChE, BChE, and total ROS with inhibition percentages (%) of 68.01, 95.72, and 73.20 at concentrations of 25 μg mL^−1^, respectively		[60]
*Ecklonia cava*	The seaweed (1 kg) was extracted with 95% ethanol (10 L) for 2 h in a water bath at 50 °C	Phlorotannins: Dieckol and phlorofucofuroeckol	Improvement of memory and possible involvement of the AChE inhibition, in vivo		[61]
	Ethanolic extract	Phlorotannin: Triphlorethol-A	Anti-oxidative activity: Scavenging activity against ROS and DPPH via activation of ERK protein, in vivo		[62]
	Methanolic extract	Phlorotnnins	In vitro scavenging activity against hydroxyl, superoxide, and peroxyl radicals IC_50_ = 392.5, 115.2 and 128.9 µM, respectively		[63]
	Enzymatic extract	Phlorotannins	The phlorotannin-rich fraction significantly potentiated the pentobarbital-induced sleep at >50 mg kg^−1^, in vivo		[64]
	80% MeOHextract	Phlorotannins	Neuroprotective effects against H_2_O_2_-induced oxidative stress in murine hippocampal HT22 cells IC_50_ = 50 µM, in vivo		[65]
	Ethanolic extract	Phloroglucinol	Reduce the toxicity ROS induced by hydrogen peroxide IC_50_ = 10 µg mL^−1^, in vivo		[66]
	Ethanolic extract	Phlorotannin: 8,8’-Bieckol	Phlorotannin: 8,8’-Bieckol reduce COX-2, NO, and prostaglandin E2 (PGE2) IC_50_ = 100 μM, in vivo		[67]
	Ethanolic extract	Extract	Extracts have potential analgesic effects in the case of postoperative pain and neuropathic pain, in vivo		[68]
	Ethanolic extract	Phlorotannin (eckol)	Inhibiting BuChE IC_50_ = 29 μM, in vitro and in vivo model studies		[69]
	Ethanolic extract	Phlorotannin (7-phloroeckol)	Inhibiting BuChE IC_50_ = 0.95 μM, in vitro and in vivo model studies		[69]
*E. kurome*	Provided by Marine Drug and FoodInstitute, Ocean University of China China	Acidic oligosaccharide sugar chain (AOSC)	Blocking the fibril formation of Aβ IC_50_ = 100 µg mL^−1^, in vitro		[70]
*E. maxima*	Crude extract was sequentially extracted with *n*-hexane, dichloromethane, ethyl acetate, and finally *n*-butanol	Phlorotannins	IC_50_ values for the solvent fractions ranged from 62.61 to 150.8 μg mL^−1^, with the ethyl acetate fraction having the best inhibitory activity, in vitro		[71]
*E. stolonifera*	Ethanolic extract	Phlorotannin (dieckol)	Inhibiting AChE 17.11 μM, in vitro		[72]
	Ethanolic extract	Phlorotannin (eckstolonol)	Inhibiting AChE and BuChE IC_50_ = 42.66 and 230.27 μM, in vitro		[72]
	Ethanolic extract	Phlorotannin (eckol)	Inhibiting AChE IC_50_ = 20.56 μM, in vitro		[72]
	Ethanolic extract	Phlorotannin (2-phloroeckol)	Inhibiting AChE IC_50_ = 38.13 μM, in vitro		[72]
	Ethanolic extract	Phlorotannin (7-phloroeckol)	Inhibiting AChE and BuChE IC_50_ = 4.89 and 136.71 μM, in vitro		[72]
	Ethanolic extract	Phlorotannin (phlorofucofuroeckol A)	Inhibiting AChE and BuChE IC_50_ = 4.89 and 136.71 μM, in vitro		[72]
	Ethanolic extract	Sterol (fucosterol)	Inhibiting AChE IC_50_ = 421.72 μM, in vitro		[72]
	Methanolic extract	Phlorotannins	Inhibiting AChE IC_50_ = 108.11 mg mL^−1^, in vitro		[73]
*Fucus vesiculosus*	Fucoidan (Sigma)	Fucoidan	Fucoidan completely blocks microglial uptake of fDNA at only 40 ng mL^−1^, in vivo		[74]
	Fucoidan (Sigma)	Fucoidan	In vitro anti-oxidative activity: Inhibit superoxide radicals, hydroxyl radicals, and lipid peroxidation IC_50_ = 0.058, 0.157 and 1.250 mg mL^−1^, respectively		[75]
	Fucoidan (Sigma)	Fucoidan	Fucoidan has protective effect via inducible nitric oxide synthase (iNOS), in vivo		[76]
	Fucoidan (Sigma)	Fucoidan	Fucoidan inhibits TNF-alpha- and IFN-gamma-stimulated NO production *via* p38 MAPK, AP-1, JAK/STAT, and IRF-1, in vivo		[77]
	Fucoidan (Sigma)	Fucoidan	Fucoidan inhibits beta amyloid induced microglial clustering at 10 µM, in vivo		[78]
	Extracted using 70% acetone	Phlorotnnins	Suppressing the overproduction of intracellular ROS induced by hydrogen peroxide IC_50_ = 0.068 mg mL^−1^, in vivo		[79]
*Ishige okamurae*	Methanolic extraction	Phlorotannin: 6,6ʹ-Bieckol	Inhibiting AChE IC_50_ = 46.42 μM, in vitro		[73]
	Methanolic extraction	Phlorotannin: Diphlorethohydroxycarmalol (DPHC)	In vivo neuroprotection against hydrogen peroxide (H_2_O_2_)-induced oxidative stress in murine hippocampal neuronal cells IC_50_ = 50 µM		[80]
*Marginariella boryana*	Sequential extractions with H_2_SO_4_ and HCl	Sulfated fucans	Prevents the accumulation of Aβ		[81]
*Padina australis*	Dichloromethane extract	Extracts	Inhibiting AChE IC_50_ = 0.149 mg mL^−1^, in vitro		[82]
*P. gymnospora*	Methanolic extract	Bioassay-guided fractionation of the active n-hexane and ethyl acetate (EtOAc) soluble fractions	Inhibiting AChE IC_50_ = 3.5 mg mL^−1^, in vitro		[72]
	Methanolic extract	Extracts	Inhibiting AChE IC_50_ = 3.5 mg mL^−1^, in vitro		[83]
	Acetone extracts	Extrats	IC_50_ value <10 μg mL^−1^ for both AChE and BuChE, in vitro		[84]
*P. tetrastromatica*	Acetone extract	Fucoxanthin	Anti-oxidative activity: Reduce lipid peroxidation in rats IC_50_ = 0.83 μM, in vivo		[85]
	Chloroform and ethanol extracts	Extract	Chloroform extract at 600 mg Kg^−1^ showed significant anticonvulsant activity, in vivo		[86]
*Papenfussiella lutea*	Sequential extractions with H_2_SO_4_ and HCl	Sesquiterpenes	Inhibiting AChE IC_50_ = 48–65 μM, in vivo		[81]
*Saccharina japonica*	Fucoidan	Fucoidan	Protective effect in MPTP-induced neurotoxicity. In addition, reduce behavioural deficits and cell death and increase dopamine IC_50_ = 25 mg kg^−1^, once per day in mice, in vivo and in vitro		[31]
	Extracted from seaweeds commercially cultured in Qingdao, China	Fucoidan	Inhibiting microglia which inhibits LPS-induced NO production via suppression of p38 and ERK phosphorylation IC_50_ = 125 μg mL^−1^, in vivo		[87]
	Fucoidan (CY110115) was obtained from Ci Yuan Biotechnology Co., Ltd., Xi’an, China. The purity of the chemical was more than 98.0%.	Fucoidan	Anti-oxidative activity: Reduce the toxicity of H_2_O_2_ in PC_12_ cells via activation of PI3K/Akt pathway IC_50_ = 60 µg mL^−1^, in vivo		[88]
	Ethanolic extract	Extracts	Promoted neurite outgrowth in a dose-dependent manner with optimal concentrations of 15 μg mL^−1^, in vitro		[89,90]
	Extracted from seaweeds commercially cultured in Qingdao, China	Fucoidan	Reduced 6-hydroxydopamine (6-OHDA) and reduced the loss of dopaminergic in neurons IC_50_ = 20 mg kg^−1^ in rats, in vivo		[91]
*Sargassum fulvellum*	MeOH-extract	Pigment: Pheophytin A	Produce neurite outgrowth (from 20 to 100% in the present of 10 ng mL^−1^ of NGF) and activate IC_50_ = 3.9 μg mL^−1^ in PC12 cells, in vivo		[92]
*S. macrocarpum*	Extracted with chloroform at room temperature	Carotenoids	Promote neurite outgrowth activity to 0.4 in PC12 cells IC_50_ = 6.25 μg mL^−1^, in vivo		[93]
	Methanol extract	Meroterpenoid: Sargaquinoic acid	Signalling pathway of TrkA-MAP kinase pathway IC_50_ = 3 μg mL^−1^, in vivo		[94]
	Methanol extract	Meroterpene: Sargachromenol	Promote survival of PC-12 cells and neurite outgrowth through activation of cAMP and MAP kinase pathways IC_50_ = 9 μM, in vivo		[95]
*S. micracanthum*	Methanol extract	Plastoquinones	Anti-oxidative activity: Lipid peroxidation IC_50_ = 0.95–44.3 μg mL^−1^ DPPH IC_50_ = 3–52.6 μg mL^-1^, in vivo		[96]
*S. fulvellum*	Ethanolic extract	Extracts	Promoted neurite outgrowth in a dose-dependent manner with optimal concentrations of 5 μg mL^−1^, in vivo		[97]
*S. fusiforme* (as *Hijikia fusiformis*)	Methanol extract	Fucoxanthin	Anti-oxidative activity: DPPH radical scavenging, in vitro		[98]
	Alcohol extract	Fucoidan	Ameliorating learning and memory deficiencies, and otential ingredient on treatment of Alzheimer’s disease, in vivo		[99]
*S. horneri*	Ethanol extract	Total sterols and β-sitosterol	Antidepressant effect, in vivo		[100]
*S. polycystum*	Hexane, dichloromethane, and methanol extracts	Extracts	Inhibiting AChE IC_50_ = 0.115, 0.180 and 0.162 mg mL^−1^, respectively, in vitro		[82]
*S. sagamianum*	MeOH extract	Sesquiterpenes	Inhibiting AChE IC_50_ = 48–65 μM, in vitro		[101]
	MeOH extract	Plastoquinones: Sargaquinoic acid and sargachromenol	Inhibiting AChE IC_50_ = 23.2 and 32.7 μM, respectively Inhibiting BuChE IC_50_ = 26 μM (for sargaquinoic), in vitro		[102]
*S. siliquastrum*	Extracted with 80% aqueous MeOH	Fucoxanthin	Anti-oxidative activity: Inhibit hydrogen peroxide in Vero cells IC_50_ = 100 uM, in vivo		[103]
	Extracted with CH_2_Cl_2_ and MeOH	Meroditerpenoids	These compounds exhibited moderate to significant radical-scavenging activity as well as weak inhibitory activities against sortase A and isocitrate lyase, in vitro		[104]
*Sargassum* sp.	Methanolic extract	Extract	Inhibiting AChE IC_50_ = 1 mg mL^−1^, in vitro		[83]
*S. swartzii* (as *Sargassum wightii*)	Extracted with CH_2_Cl_2_ and MeOH	Alginic acid	Polysaccharide inhibition activities to COX-2, lipoxygenase (5-LOX), xanthine oxidase (XO) and myeloperoxidase (MPO) in type II collagen induced arthritic rats IC_50_ = 100 mg kg^−1^, in vivo		[105]
	Petroleum ether, hexane, benzene, and dichloromethane extracts	Extracts	Inhibiting AChE IC_50_ = 19.33, 46.81, 27.24, 50.56 µg mL^−1^, respectively Inhibiting BuChE IC_50_ = 17.91, 32.75, 12.98, 36.16 µg mL^−1^, respectively, in vivo		[106]
*S. vulgare*	Methanolic extract	Extracts	Inhibiting AChE IC_50_ = 3.5 mg mL^−1^, in vitro		[54]
*Scytothamnus australis*	6 h with 1% (**w/v**) H_2_SO_4_ at 20 °C, 6 h with 0.2 M HCl at 20 °C, 6 h with 2% CaCl_2_ at 75 °C	Sulfated fucans	Prevents the accumulation of Aβ, in vivo		[81]
*Splachnidium rugosum*	6 h with 1% (**w/v**) H_2_SO_4_ at 20 °C, 6 h with 0.2 M HCl at 20 °C, 6 h with 2% CaCl_2_ at 75 °C	Sulfated fucans	Prevents the accumulation of Aβ, in vivo		[81]
*Turbinaria decurrens*	Dried seaweed powder was depig-mented with acetone followed by hot water extraction at 90–95 °C for 3–4 h	Fucoidan	Potential neuroprotective effect in Parkinson’s deasese, in vivo		[107]
*Undaria pinnatifida*	Ethanolic extract	Extract	Promoted neurite outgrowth in a dose-dependent manner with optimal concentrations of 5 μg mL^−1^, in vitro		[90,91]
	Ethanolic extract	Extract	Neurogenesis, neuroprotection, anti-inflammatory and anti-Alzheimer’s, in vivo		[108]
	Glycoprotein	Glycoprotein	Neurogenesis, neuroprotection, anti-inflammatory and anti-Alzheimer’s Showed predominantly AChE, BChE, and BACE1 inhibitory activities with IC_50_ values of 63.56, 99.03 and 73.35 μg mL^−1^, respectively, in vitro and in vivo		[109]
	Ethanolic extract	Sulfated fucans	Prevents the accumulation of Aβ, in vivo		[81]
*Zonaria spiralis*	Ethanolic extract	Phloroglucinol: Spiralisone A and Chromone 6	Kinases inhibitory to CDK5/p25, CK1δ and GSK3β IC_50_ = 10.0, <10 and <10 μM, respectively, in vitro		[38]
Rhodophyta (red seaweeds)					
*Amphiroa beauvoisii*	50% Aqueous methanol extract	Phenolic, flavonoid extracts	Inhibiting AChE IC_50_ = 0.12 mg mL^−1^, in vitro	[110]	
*A. bowerbankii*	Methanolic extract	Extracts	Inhibiting AChE IC_50_ = 5.3 mg mL^−1^, in vitro	[56]	
*A. ephedraea*	Methanolic extract	Extracts	Inhibiting AChE IC_50_ = 5.1 mg mL^−1^, in vitro	[56]	
*Asparagopsis armata*	Methanolic extract	Extracts	AChE inhibitory capacity: 58.4% BuChE inhibitory capacity: 81.4% 10 mg mL^−1^, in vitro	[54]	
*Bryothamnion triquetrum*	Water extract	Fractions	Protect GT1–7 cells death produced by severe (180 min.) chemical hypoxia/aglycemia insult, in vitro	[21,111]	
*Chondracanthus acicularis*	Alcaline extraction	Lambda-carrageenan	Anti-oxidative activity: Inhibit superoxide radicals, hydroxyl radicals and lipid peroxidation IC_50_ = 0.046, 0.357 and 2.267 mg mL^−1^, respectively, in vitro	[75]	
*Chondrophycus undulatus* (as *Laurencia undulata*)	Glycerol glycosides: Floridoside	Glycerol glycosides: Floridoside	Suppress pro-inflammatory responses in microglia by markedly inhibiting the production of nitric oxide (NO) and reactive oxygen species (ROS) IC_50_ = 10 μM	[112]	
*Chondrus crispus*	Methanolic extract	Floridoside and d-Isofloridoside	Extract-mediated protection against Parkinson’s disease pathology, in vitro and in vivo	[47]	
*Eucheuma denticulatum*	Alcaline extraction	Iota-carrageenan	Anti-oxidative activity: Inhibit superoxide radicals, hydroxyl radicals and lipid peroxidation IC_50_ = 0.332, 0.281 and 0.830 mg mL^−1^, respectively, in vitro	[75]	
*Gelidiella acerosa*	Petroleum ether, hexane, benzene, dichloromethane, chloroform, ethyl acetate, acetone, methanol, and water extracts	Extracts	Inhibiting AChE Benzene extract, IC_50_ = 434.61 μg mL^−1^ Ethyl acetate, IC_50_ = 444.44 μg mL^−1^ Inhibiting BuChE Benzene extract, IC_50_ = 163.01 μg mL^−1^ Chloroform extract, IC_50_ = 375 μg mL^−1^, in vitro	[113]	
	Petroleum ether and successively extracted with benzene	Phytol	In vitro and in vivo antioxidant activities (25–125 μg mL^−1^) with an IC_50_ value of 95.27 μg mL^−1^ and cholinesterase inhibitory potential (5–25 μg mL^−1^) with IC_50_ values of 2.704 and 5.798 μg mL^−1^ for AChE and BuChE, respectively, in vitro	[114]	
*Gelidium amansii*	Ethanol extract	Extract	Neurogenesis (synaptogenesis promotion), in vitro and in vivo	[90,115]	
*G. foliaceum*	50% Aqueous methanol extract	Phenolic and Flavonoid compouds	Inhibiting AChE IC_50_ = 0.16 mg mL^−1^, in vitro	[110]	
*Gloiopeltis furcata*	2-(3-Hydroxy-5-oxotetrahydrofuran-3-yl) acetic acid, glutaric acid, succinic acid, nicotinic acid, (E)-4-hydroxyhex-2-enoic acid, cholesterol, 7-hydroxycholesterol, uridine, glycerol, phlorotannin, fatty acids		Inhibiting AChE 1.4–12.50 μg mL^−1^ Inhibiting BuChE 6.56–75.25 μg mL^−1^, in vitro	[116]	
*Hydropuntia edulis* (as *Gracilaria edulis)*	Methanolic extract	Extract	Inhibiting AChE IC_50_ = 3 mg mL^−1^, in vitro	[83]	
	Methanolic extract	Extract	Inhibiting AChE IC_50_ = 3 mg mL^−1^, in vitro	[72]	
*Gracilaria gracilis*	Methanolic extract	Extract	Inhibiting AChE IC_50_ = 1.5 mg mL^−1^, in vitro	[83]	
*Gracilariopsis chorda*	Ethanolic extract	Extract	Neuronal cell viability and cell cytotoxicity testing revealed that the ethanol extract afforded the most neuroprotection at a concentration of 15 µmL^−1^, at which G. chorda extract significantly increased cell viability to 119.0–3.2%, and decreased cell death to 80.5–10.3%, in vivo	[46]	
*G. chorda*	Ethanolic extract	Extract	Extract concentration-dependently increased neurite outgrowth, with an optimal concentration of 30 mu g mL^−1^, in vivo	[117]	
*Hypnea valentine*	Methanolic extract	Extract	Inhibiting AChE IC_50_ = 2.6 mg mL^−1^, in vitro	[83]	
	Methanolic extract	Extracts	Inhibiting AChE IC_50_ = 2.6 mg mL^−1^, in vitro	[118]	
*Kappaphycus alvarezii*	Alcaline extraction	Kappa-carrageenan	Anti-oxidative activity: Inhibit superoxide radicals, hydroxyl radicals and lipid peroxidation IC_50_ = 0.112, 0.335 and 0.323 mg mL^−1^, respectively, in vitro	[75]	
	Ethanolic extract	Extracts	Promotes neurite outgrowth in hippocampal neurons, in vivo	[119]	
*Ochtodes secundiramea*	Dichloromethane/methanol extract	Halogenated monoterpenes	Extract showed 48% AChE inhibition at 400 µg mL^−1^, in vitro	[120]	
*Porphyra/Pyropia* sp. (Korean purple laver)	In vitro digestion	Phycoerythrobilin	Antioxidant activity IC_50_ = 0.048 mmol g^−1^, in vitro	[121]	
*Neopyropia yezoensis* (as *Porphyra yezoensis*)	Ethanolic extract	Extract	Increased neurite outgrowth at an optimal concentration of 15 µg mL^−1^, in vivo	[122]	
*Rhodomela confervoides*	Ethyl acetate extract	Bromophenols	Antioxidant activity IC_50_ = 5.22–23.60 µM, in vitro	[123]	
*Rhodomelopsis africana*	50% aqueous methanol extract	Phenolic and Flavonoid compouds	Inhibiting AChE IC_50_ = 0.12 mg mL^−1^, in vitro	[110]	
**Chlorophyta (green seaweeds)**					
*Caulerpa racemosa*	Methanolic extract	Extract	Inhibiting AChE IC_50_ = 5.5 mg mL^−1^, in vitro		[56]
	Extracted with MeOH and partitioned between H_2_O and hexane, chloroform, ethyl acetate and n-butanol.	Alkaloid: Caulerpin	Inhibition of nociception 100 μM kg^−1^ in Swiss albino mice, in vivo		[124]
	Ethanol extract	Bisindole alkaloid (racemosin A)	Increase 5.5% of cell viability in SH-SY5Y cells (neuroblast from neural tissue) IC_50_ = 10 µM, in vivo		[125]
	Ethanol extract	Bisindole alkaloid (racemosin B)	Increase 14.6% of cell viability in SH-SY5Y cells (neuroblast from neural tissue) IC_50_ = 10 µM, in vivo		[125]
	Hexane, dichloromethane and methanol extracts	Extracts	Inhibiting AChE IC_50_ = 0.086, 0.089 and 0.095 mg mL^−1^, respectively Inhibiting BuChE IC_50_ = 0.156, >0.2 and 0.118 mg mL^−1^, respectively, in vitro		[82]
	Ethanol extract	Terpenoid (α-tocospirone)	13.85% increases in cell viability in SH-SY5Y cells IC_50_ = 10 μM, in vivo		[126]
	Ethanol extract	Sterol (23E)-3β-hydroxystigmasta-5,23-dien28-one	11.31% increases in cell viability in SH-SY5Y cells IC_50_ = 10 μM, in vivo		[126]
	Ethanol extract	Sterol (22E)-3β-hydroxycholesta-5,22-dien24-one	15.98% increases in cell viability in SH-SY5Y cells IC_50_ = 10 μM, in vivo		[126]
*Cladophora vagabunda* (as *Cladophora fascicularis*)	Methanolic extract	Extract	Inhibiting AChE IC_50_ = 2 mg mL^−1^, in vitro		[83]
*Codium capitatum*	Methanolic extract	Extract	Inhibiting AChE IC_50_ = 7.8 mg mL^−1^, in vitro		[56]
	50% Aqueous methanol extract	Phenolic and Flavonoid compouds	Inhibiting AChE IC_50_ = 0.11 mg mL^−1^, in vitro		[110]
*C. duthieae*	50% Aqueous methanol extract	Phenolic and Flavonoid compouds	Inhibiting AChE IC_50_ = 0.14 mg mL^−1^, in vitro		[110]
*C. fragile*	80% aqueous methanol extract	Sterol: Clerosterol	Exhibit reducing activity to COX-2, iNOS and TNF-α IC_50_ = 3 μg mL^−1^, in vitro and in vivo		[127]
*Halimeda incrassata*	Water extracts	Extracts	Neuroprotective and antioxidant properties, in vitro and in vivo		[21]
*H. cuneata*	Methanolic extract	Extracts	Inhibiting AChE IC_50_ = 5.7 mg mL^−1^, in vitro		[56]
	50% Aqueous methanol extract	Phenolic and Flavonoid compouds	Inhibiting AChE IC_50_ = 0.07 mg mL^-1^, in vitro		[110]
*Ulva australis* (as *Ulva pertusa*)	Water at 125 °C for 4 h; polysaccharides were precipitated by the addition of 4000 mL of 95% (**v/v**) ethanol	Sulfated polysaccharide (ulvan)	Scavenging activity for superoxide radicals, in vitro		[128]
*U. fasciata*	Methanolic extract	Extracts	Inhibiting AChE IC_50_ = 4.8 mg mL^−1^, in vitro		[56]
	50% Aqueous methanol extract	Phenolic and Flavonoid compouds	Inhibiting AChE IC_50_ = 0.13 mg mL^−1^, in vitro		[110]
*U. prolifera* (as *Enteromorpha prolifera*)	95% ethanol extract	Pheophorbide A	Antioxidant activity IC_50_ = 71.9 µM, in vitro		[129]
*U. reticulata*	Methanolic extract	Extract	Inhibiting AChE IC_50_ = 10 mg mL^−1^, in vitro		[83]
	Methanolic extract	Extract	Inhibiting AChE IC_50_ = 10 mg mL^−1^, in vitro		[118]

## 4. Multiple Sclerosis, Other Chronic Diseases, and the Seaweed Diet

Multiple Sclerosis (MS) is the most common of the chronic, inflammatory, neurological demyelination diseases of the CNS, of autoimmune origin. Although the cause of the disease is still unknown, MS has been the focus of many studies worldwide, which has made possible a constant and significant evolution in patients’ quality of life. Patients are usually young, especially women in their 20s and 40s [130,131]. MS cannot be cured and can be manifested by various symptoms such as: Severe fatigue, depression, muscle weakness, impaired balance of motor coordination, joint pain, bowel and bladder dysfunction, and changes in visual acuity [131].

Both modifiable lifestyle factors and the quality of the diet can affect the course of the disease, dietary guidelines for people with MS have the potential to reduce symptoms related to the disease. Potential mechanisms by which the quality of the diet can influence the course of the disease in patients with MS include epigenetic changes in gene expression and changes in the composition of the intestinal microbiome, which can result in reduced inflammation. The quality of the diet can also influence the sufficiency of nutrients needed for neuronal structure [132].

There is currently no formal recommendation for the application of a specific dietary protocol for MS, although evidence has accumulated favoring the implementation of low-fat diets. The first of these diets was developed by Roy Swank in the 1950s, characterized by a low amount of saturated fats, not exceeding 15 g, and by supplementation with cod liver oil [50]. A modified version of this diet, the McDougall diet, has been studied and has recently shown to be useful in the fight against MS-related fatigue [133].

The synthesis of poly-unsaturated fatty acids (PUFAs), omega 3 (ω-3 fat acids), and 6 (ω-6 fat acids) is made by vegetables. In marine algae, despite their low lipid content, these fatty acids may form a significant part of the lipid profile of algae. Thus, PUFAs are important components of cell membranes and precursors of eicosanoids, essential bio-regulators of various cellular processes. PUFAs, effectively, reduce the risk of cancer, diabetes, hypercholesterolemia, cardiovascular diseases, osteoporosis, and MS progression. As there is historical and recurrent use of seaweed in Asia, in addition to increasing food use in other parts of the world, there is great potential for an increase in the presence of ω-3 PUFAs, especially in the western diets [134,135,136,137]. 

The major commercial sources of ω-3 PUFAs are fish, but their wide usage as food additives is limited because of the typical “fishy smell”, unpleasant taste, and lack of stability (since they are powerful antioxidants) [138]. The term “a balanced diet” of a heterotrophic organism implies the intake of essential nutrients for growth and reproduction. Some of the essential nutrients may be used for reconstruction, decomposed, and/or used in the production of new metabolites essential to the primary metabolism. However, there are others which cannot be produced, requiring them to be obtained externally through ingestion. Among these are poly-unsaturated fats, of which the most familiar are the fatty acids of omega type, as we have seen previously. These fatty acids control the cholesterol that binds to lipoproteins (carriers of these fats in blood plasma), that is, the balance between HDL-C (High Density Lipoprotein Cholesterol), or good cholesterol, and LDL-C (Low Density Lipoprotein Cholesterol), the bad cholesterol. The first should be kept at a high rate, the second at a low rate. HDL-C carries excess cholesterol into the bloodstream to the liver, where it is catabolized, while LDL-C does reverse transport, thus promoting its accumulation in tissues and organs [139]. EPA (eicosapentaenoic acid) and DHA (docosahexaenoic acid) are carboxylic acids of omega-3 type, which are considered the most important polyunsaturated fatty acids for human health (220 mg daily), because the human body is unable to produce them, and only obtains them through the intake of foods that contain them. Amongst the main sources of this type of unsaturated fats are, in addition to algae, the fish from the cold deep waters (sardines, salmon, trout, tuna, and mackerels, etc.). There is also another family of polyunsaturated fatty acids, the omega-6 group, whose primary sources are, in addition to the algae, vegetable oils (soy, corn, sunflower, etc.), fats, and eggs. Curiously, in the human body, both groups of these polyunsaturated fatty acids are reported to interact, i.e., so that omega-3 can act effectively give rise to all its potential benefits, there needs to be a balance between the consumption of omega-3 and omega-6 in the diet [139].

Omega-3 is the only fatty acid which can lower triglycerides (i.e., decreases hepatic synthesis of these fats) and hence the conversion of LDL-C in the liver—which is thus available to be transported to various tissues, which can then be deposited, thereby increasing the likelihood of some diseases. Accordingly, high levels of EPA may play an important role in the prevention of thrombotic strokes, and may also lower the risk of atherosclerosis, ischemic heart disease or angina pectoris, and the risk of myocardial stroke [139,140]. 

For example, the red seaweed Dulse (*Palmaria palmata*) (Figure 2i), and the brown seaweeds Kombu (*Saccharina latissima*) (Figure 2p) and *Saccorhiza polyschides* (Figure 2q), beyond the polyphenols, vitamins A, B_12_, and C (see Table 2), high levels of fiber, protein, minerals, and arginine, has low concentration of saturated fatty acids, and various polyunsaturated fatty acids—such as linoleic and arachidonic of omega-6 family, or EPA omega-3 family. It is therefore a good seaweed for restorative in anemia states, asthenia (weakness), and for postoperative processes. It also strengthens the vision (has high levels of vitamin A) and is recommended for the treatment of gastric and intestinal disorders, and for the regeneration of mucosa (respiratory, gastric, and vaginal). Due to the high content of vitamin B_12_, this red seaweed is suggested to provide a protection against cardiovascular diseases [141,142]; and because this vitamin also reduces the homocysteine levels in blood, when high amounts are deposited in blood vessels [136,138].

Phlorotannins (brown algae, such as *Fucus vesiculosus* and *Ascophyllum nodosum,* polyphenols) are structural analogues of condensed tannins from terrestrial plants and are found exclusively in algae and macrophytes. Phlorotannins are polymers made up of phloroglucinol (1,3,5-trihydroxybenzene) and can make up 25% of the dry algae biomass. They exhibit anti-inflammatory, antioxidant, antibacterial, antiallergic, and anti-tumor properties, and inhibit antiplasmins, matrix metalloproteinases, etc. [143].

According to Valado et al. [144], the daily intake of vegetable jelly for 60 days showed a reduction in serum total cholesterol (TC) and LDL-C levels in women, leading to the conclusion that carrageenan has bioactive potential in reducing TC concentration.

The results obtained in the work of Nunes et al. [145] strongly reinforce the possibility of using *Ulva lactuca* (Figure 2o) and *Zonaria tournefortii* (Phaeophyceae) lipids to improve the nutraceutical characteristics of food products and develop supplements based on macroalgae to improve or maintain health. Considering the above, including *U. lactuca* and *Z. tournefortii* in human nutrition can be a means of reducing the ω6/ω3 fat acids ratio.

In addition to controlling the amount of saturated fats in the diet, nutritional conduits like non-MS subjects are recommended, according to the patient’s biochemical individuality and current recommendations. Food quality should be prioritized in order to maintain or restore patients’ nutritional status and avoid chronic diseases such as obesity, diabetes mellitus type 2, systemic arterial hypertension, dyslipidemias, and cardiovascular diseases [134,137].

Although poorly understood and studied, dietary habits and the nutritional status of individuals with MS suggest that they may suffer from different nutritional imbalances. Among these, we can highlight obesity (more common), cachexia, low weight, and vitamin deficiencies. If they occur, vitamin deficiencies may interfere with the proper functioning of the immune system and the symptoms of the disease, the most important being vitamins A and D (modulators of the immune system), C, E, and B_12_ (important for myelin synthesis) (see Table 2) [146].

Once this relationship was established, studies in animal models have confirmed the neuroprotective properties of vitamin D, opening precedents for studies in humans [147]. One of the most important studies in the field [148] showed that the supplementation of 400 IU daily of vitamin D was able to reduce the number of relapsing, seeming to reduce the severity of the disease.

Vitamin D (Calciferol), another fat-soluble vitamin, whose functions are maintaining phosphorus and calcium concentrations in the blood, regulate bone metabolism and promote calcium fixation in bones and teeth; in children, it is key to bone growth [4,23].

Seaweeds, when included in a varied and balanced diet, can contribute effectively to provide adequate intake of a wide variety of vitamins (Table 2). Vitamins can be divided into those that are soluble in water, and those that are liposoluble. Water-soluble vitamins include B-complex vitamins and vitamin C. The B-complex vitamins are the largest group, and perform activities associated with muscle tone, metabolism, cell growth, and nervous system. For example, Sea Lettuce (*Ulva* spp., Figure 2j) (green algae) and Nori (*Neopyropia/Pyropia/Porphyra* spp., Figure 2k,l) (red algae) are excellent sources of vitamin B_12_ (Cobalamin), which is essential in DNA synthesis. Vitamin C (Ascorbic acid) is a water-soluble vitamin very important for oral health (gums), iron absorption, and resistance of the body to infections [4,23].

Vitamins A, D, E, and K are all fat-soluble. The liver is the major organ responsible for the storage, metabolism, and distribution of Vitamin A (Retinol) to peripheral tissues [149]. It is also suggested that the liver, besides functioning as a deposit site for vitamin A, can use retinol for its normal functioning, such as the proliferation and differentiation of its cells [150]. Vitamin A plays a key role in maintaining the integrity of visual processes, as an inadequate condition is the leading cause of preventable blindness in childhood. Another important nutritional role of vitamin A is its involvement in the immune system [151].

Vitamins E (Tocopherol) and K also have several biological functions, including antioxidant activity and blood clotting. In addition to its biochemical functions and antioxidant activity, vitamins derived from seaweed have other health benefits, such as: Preventing cardiovascular diseases; reducing hypertension; and reducing the risk of cancer [152,153,154].

The genus and species to which the macroalgae belong is a critical factor that can affect the vitamin composition. For example, the level of Niacin (vitamin B_3_) in some Phaeophyceae (*Laminaria* spp., Figure 2a) is approximately one tenth of the level found in Rhodophyta, *Neopyropia tenera* (as *Porphyra tenera*). Other factors that may influence vitamin content include geographic location, water temperature, salinity, and harvest time. Vitamin content may also be affected by processing, as both the dehydration and the temperature at which macroalgae are subjected may affect the levels of vitamin present [153,154].

## 5. Conclusions

The stress of our day-to-day lives and lack of time often lead us to opt for fast food, rich in calories and saturated fats, which leads to a lack of essential nutrients, obesity, and the appearance of diseases related to an excessive intake of sugars and fats, such as diabetes and arteriosclerosis, among others.

Algae are a natural food that provides us with a high nutritional value, but low in calories. With very low values in fats, marine algae have polysaccharides that behave, for the most part, as non-caloric fibers. Algae provide a rich and healthy diet (in trace elements and vitamins), offering a multitude of flavors, aromas, and textures, becoming an alternative food that easily awakens the curiosity of the most curious palates. Because they are rich in minerals, vitamins, and fiber, but low in lipids, they are undoubtedly an excellent choice for weight loss regimens and can even facilitate intestinal transit, lower blood cholesterol levels, and reduce certain affections such as colon cancer, and helps to delay, or at least minimize, the effects of neurodegenerative diseases, such as AD, PD, MS, among others.

Several in vitro studies have proven the efficacy of nutraceuticals and food products fortified from bioactive compounds from seaweed, however, the most challenging factor in the food industry is to develop new products that can attract consumers where strange or unfamiliar products are approaching them.

## Figures and Tables

**Figure 1 marinedrugs-19-00128-f001:**
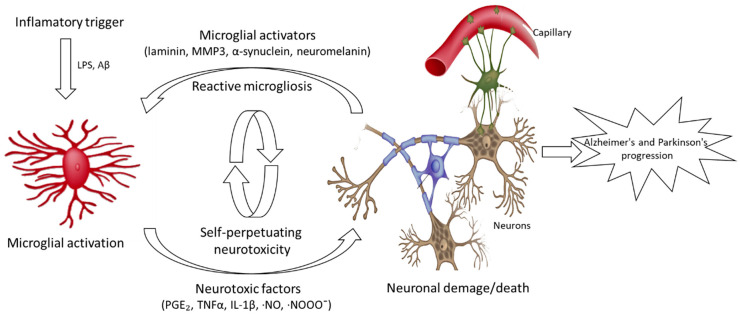
Microglia-mediated neurotoxicity in Alzheimer’s and Parkinson’s diseases.

**Figure 2 marinedrugs-19-00128-f002:**
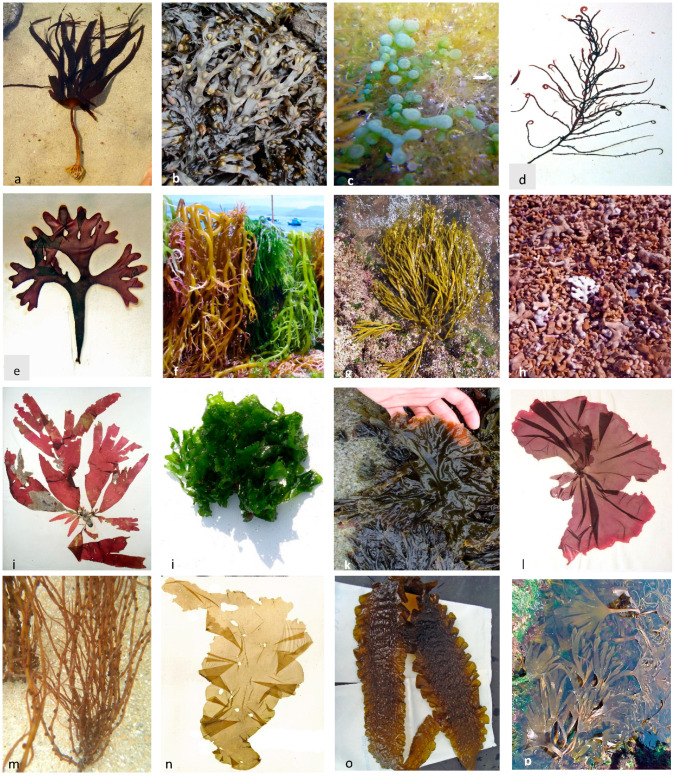
Some seaweeds with neurophysiological activities: (**a**)—Laminaria hyperborea (B); (**b**)—Fucus vesiculosus (B); (**c**)—Caulerpa racemosa (G); (**d**)—Hypnea musciformis (R); (**e**)—Chondrus crispus (G); (**f**)—Kappaphycus alvarezii (R); (**g**)—Bifurcaria bifurcata (B); (**h**)—Lithothamnion corallioides (R); (**i**)—Palmaria palmata (R); (**j**)—Ulva rigida (G); (**k**)—Neopyropia leucosticta (R); (**l**)—Porphyra umbilicalis (R); (**m**)—Gracilaria gracilis (R); (**n**)—Ulva lactuca (G); (**o**)—Saccharina latissima (B); (**p**)—Saccorhiza polyschides. B: Brown Algae; G: Green Algae; R: Red Algae.

**Table 2 marinedrugs-19-00128-t002:** Vitamin content of some edible seaweeds (mg/100 g edible portion).

Species/Vitamins	A	B_1_	B_2_	B_3_	B_5_	B_6_	B_8_	B_12_	C	D	E	Folic Acid	References
**Chlorophyta**													
*Ulva lactuca*	0.017	<0.024	0.533	98 *	-	-	-	6 *	<0.242	-	-	-	[155,156]
*U. rigida*	9581	0.47	0.199	<0.5	1.70	<0.1	0.012	6	9.42	-	19.7	0.108	[157]
**Phaeophyceae**													
*Alaria esculenta*	-	0.6	0.3–2.7	11	-	6	-	5	100–500 *	-	-	-	[156,158]
*Ascophyllum nodosum*	-	1.5	0.6	1.9	-	<0.1	-	<0.1	52	-	-	15	[158]
*Fucus vesiculosus*	0.307	0.02	0.035	-	-	-	-	-	14.124	-	-	-	[155,159]
*Himanthalia elongata*	0.079–0.3	0.020–0.3	0.020–4.5	-	-	-	-	-	28.56–66	-	5.8	0. 176–0.258	[155,158,159,160]
*Laminaria digitata*	-	0.3–1.250	0.138–0.8	2.6–6.12	-	6.41	6.41	0.0005	16–35.5	-	3.43–4.7	-	[157,161]
*L. ochroleuca*	0.041	0.058	0.212	-	-	-	-	-	0.353	-	-	0.479	[155,160]
*Lobophora variegata*	10.340	0.3771	0.3491	4.0162	1.36	0.3040	-	0.119	23.430	0.6442	2.13	1.983	[162]
*Saccharina japonica*	0.481	0.2	0.85	1.58		0.09	-	-	-		-	-	[163]
*S. latissima*	0.04–0.4	0.05–0.2	0.21–0.4	1.7	-	0.2	-	0.0003-0.2	0.35	18	1.6		[158,159]
*Undaria pinnatifida*	0.04–0.22	0.17–0.30	0.23–1.4	2.56	-	0.18	-	0.0036	5.29		1.4–2.5	0.479	[155,159,160,163]
**Rhodophyta**													
*Chondrus crispus*	<0.1	<0.1	2.5	3.2	-	0.4	-	0.6–4 *	10–13 *	16	-	4.7	[156,158,164]
*Gracilaria* spp.	-	-	-	-	-	-	-	-	16–149 **	-	-	-	[165]
*Palmaria palmata*	1.59–3.7	0.073–1.56	0.51–1.91	1.89–2.6	-	6.8–8.99	-	0.009–3.5	6.34–34.5	-	2.2–13.9	0.267–3.5	[156,158,159,160]
*Porphyra umbilicalis*	3.65	0.144	0.36	-	-	-	-	0.029	4.214	60	-	0.363	[155,159,160]
*Neopyropia yezoensis*	16000 ***	0.129	0.382	11.0	-	-	-	0.052	-	-	-	-	[166,167]

* Expressed as ppm; ** expressed as mg%; *** expressed as I.U.; A = Retinol; B_1_ = Thiamin; B_2_ = Riboflavin; B_3_ = Niacin; B_5_ = Pantothenic Acid; B_6_ = Pyridoxine; B_8_ = Biotin; B_12_ = Cobalamin; C = Ascorbic Acid; D = Cholecalciferol; E = Folic Acid.

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
