# Peer review of "The Seaweed Diet in Prevention and Treatment of the Neurodegenerative Diseases"

_marinedrugs, 2021, doi:10.3390/md19030128_

Round 1
Reviewer 1 Report
Title: The Seaweed Diet in Prevention and Treatment of the Neurodegenerative and Other Chronic Diseases.
Personally, I think the article is very interesting. I appreciate the effort made by the authors to develop a diet that includes algae products or derivatives of them to prevent diseases of great impact in today's society such as neurodegenerative and chronic diseases.
The diseases treated in this study are all neurodegenerative, therefore it would be advisable to change the title. It is true that they are also chronic diseases, but the fact of saying in the title neurodegenerative diseases and other chronic diseases makes one expect a section with these other diseases. My recommendation is to remove that part of the title. Other option it to add a section with diseases such as diabetes, cancer, asthma, among others. For this last option, it would also be necessary to change the introduction section. Modifying the introduction would be a shame because it seems excellent to me.
In section 2, some abbreviations are introduced without being explained beforehand. It is necessary to explain what they mean. In addition, it would be convenient to make a table that collect the abbreviations since quite a few appear in the text. In this way its understanding would be facilitated.
In section 3, the scientific names of the species should be in italics. Correct it in all the manuscript. Also, this section contains a lot of information but it appears disorganized. It does not have a common thread. It would also be convenient to create a table with the different studies consulted, the results obtained, the applications, etc.
Section 4 is interesting, but it has little focus on algae. Much information is given, but a low percentage of it is dedicated to algae. Furthermore, few examples are given of studies in which algae have been used for such purposes.
The photos have a different format. In some you can see the alga with a scale to determine its size and in others you cannot. Unify the format. Either all with the scale, or none at all.
There is a table of the vitamin content of algae, but no information is given on any other compound present in them. Why so important in this table? Why is this done for vitamins and not for example for sugars that have been shown to have antineurodegenerative activity? That table is the object of study and should be made.
The title of the article gives great importance to neurodegenerative diseases and this is so until the middle of the article. Later these diseases lose importance and are hardly mentioned in the conclusions. Reassess what the article wants to be to focus it in a better way.
FINAL REMARKS
In my opinion, authors have carried out a really interesting study, with promising expectations for future research. The manuscript is clear and well written. However, I am suggesting MAYOR REVISIONS. The study should be improved before publication.
Author Response
Personally, I think the article is very interesting. I appreciate the effort made by the authors to develop a diet that includes algae products or derivatives of them to prevent diseases of great impact in today's society such as neurodegenerative and chronic diseases.
Answer: Thank you very much for the review and comments made. We will proceed with the suggested corrections. All corrections and changes were made in track changes mode
The diseases treated in this study are all neurodegenerative, therefore it would be advisable to change the title. It is true that they are also chronic diseases, but the fact of saying in the title neurodegenerative diseases and other chronic diseases makes one expect a section with these other diseases. My recommendation is to remove that part of the title. Other option it to add a section with diseases such as diabetes, cancer, asthma, among others. For this last option, it would also be necessary to change the introduction section. Modifying the introduction would be a shame because it seems excellent to me.
Answer: Yes, in fact much of the manuscript is about the usefulness of food based on seaweed and its compounds in neurodegenerative diseases, so we agree to change the title as suggested
In section 2, some abbreviations are introduced without being explained beforehand. It is necessary to explain what they mean. In addition, it would be convenient to make a table that collect the abbreviations since quite a few appear in the text. In this way its understanding would be facilitated.
Answer: According th author instructions, abbreviations should be defined in parentheses the first time they appear in the abstract, main text, and in figure or table captions and used consistently thereafter. We made a check of all the abbreviations that appear in the text, so that the first time it appears it is explained in full
In section 3, the scientific names of the species should be in italics. Correct it in all the manuscript. Also, this section contains a lot of information but it appears disorganized. It does not have a common thread. It would also be convenient to create a table with the different studies consulted, the results obtained, the applications, etc.
Answer: The algae scientific names have all been corrected. The strategy used to present the information, and since it is very vast, was to present the different studies in ascending chronological order. As the method of presenting the bibliography cited in Marine Drugs is by numbers and not by author/year, this strategy is not noticeable! We accepted the suggestion to add a table so that the information is more condensed.
Section 4 is interesting, but it has little focus on algae. Much information is given, but a low percentage of it is dedicated to algae. Furthermore, few examples are given of studies in which algae have been used for such purposes.
Answer: More information was added to this section as the summarization of compounds with bioactivities related to neurodegenerative diseases was made in Table 1
The photos have a different format. In some you can see the alga with a scale to determine its size and in others you cannot. Unify the format. Either all with the scale, or none at all.
Answer: The scales were removed in order to standardize all images
There is a table of the vitamin content of algae, but no information is given on any other compound present in them. Why so important in this table? Why is this done for vitamins and not for example for sugars that have been shown to have antineurodegenerative activity? That table is the object of study and should be made.
Answer: Moreover, we will add an additional table, organized according the different taxonomic groups of algae (brown, red and green - see Table 1)
The title of the article gives great importance to neurodegenerative diseases and this is so until the middle of the article. Later these diseases lose importance and are hardly mentioned in the conclusions. Reassess what the article wants to be to focus it in a better way.
Answer: done
FINAL REMARKS
In my opinion, authors have carried out a really interesting study, with promising expectations for future research. The manuscript is clear and well written. However, I am suggesting MAYOR REVISIONS. The study should be improved before publication
Authors: All corrections and changes were made in track changes mode
Reviewer 2 Report
Algae are rich sources of various groups of compounds, including carbohydrates, fats, vitamins, amino acids, carotenoids, trace elements, and polyphenols. Please structure the manuscript from point of view of the effect of different groups of compounds on neurodegenerative diseases.
The use of plants for the treatment of aging‐related diseases is known (e.g. https://doi.org/10.1002/med.21743). What is the advantage of the algae diet in comparison with the one considered by the authors. Please discuss this aspect.
What is the advantage of using vitamins from the seaweed diet compared to synthetic vitamins. Consider the effects of carotenoids, including fucoxanthin, from algae on aging‐related disease.
Recently, algae polyphenols have received more attention. Please discuss the accumulation of phlorotanins in algae (e.g. https://doi.org/10.1007/s11094-019-01987-0) and their effect on neurodegenerative diseases.
A key aspect of the diet is the absorption of its compounds. Algae contain a large amount of carbohydrates. Fucoidan is the main component of brown algae with a broad spectrum of action, the effect of which depends on various factors, including molecular weight (e.g. https://doi.org/10.3390/ijms21239272). Consider the problem of preventing neurodegenerative diseases in terms of the absorption of fucoidan (for example, https://doi.org/10.3390/md16040132).
The use of algal polysaccharides for the treatment of neurodegenerative diseases has been widely considered. Discuss your details with ( e.g. https://doi.org/10.3390/md19020089).
Please discuss the aspects of the use and absorption of other carbohydrates, namely lamiranan, carriginan and others (for example, https://doi.org/10.3390/md18110557).
For a successful visualization of the manuscript data, provide the data in the form of a table on the use of the algal diet in humans and animals. Compare data from humans and animals.
The manuscript was written by various authors. Please reformat so that it is a single text.
Author Response
Answer: Thank you very much for the review and comments made. We will proceed with the suggested corrections. All corrections and changes were made in track changes mode.
Algae are rich sources of various groups of compounds, including carbohydrates, fats, vitamins, amino acids, carotenoids, trace elements, and polyphenols. Please structure the manuscript from point of view of the effect of different groups of compounds on neurodegenerative diseases.
Answer: The strategy of this review is to emphasize the use of algae and its compounds in some pathologies related to the nervous system diseases, and the organization decided by the authors is of diseases and not the groups of compounds. However, we will consider the comments and suggestions of the reviewer, adding the articles and groups of compounds suggested. Additionally, we will add an additional table, organized according the different taxonomic groups of algae (brown, red and green - see Table 1)
The use of plants for the treatment of aging‐related diseases is known (e.g. https://doi.org/10.1002/med.21743). What is the advantage of the algae diet in comparison with the one considered by the authors. Please discuss this aspect.
Answer: This paragraph was added in the introduction: “Adaptogens comprise a category of medicinal and nutritional products based on plants that promote adaptability, resilience and survival of living organisms under stress. Common adaptogenic plants used in various traditional medical systems (TMS) and conventional medicine and provide a modern justification for their use in the treatment of stress-induced and age-related diseases”
What is the advantage of using vitamins from the seaweed diet compared to synthetic vitamins.
Answer: The second paragraph of the introduction refers to the advantage of using compounds of natural origin over synthetic ones, including vitamins: “However, it is believed that synthetic neuroprotective agents may have side effects, such as tiredness, drowsiness, numbness in the upper and lower limbs, balance difficulties, nervousness, or anxiety, etc.”
Consider the effects of carotenoids, including fucoxanthin, from algae on aging‐related disease.
Answer: The effect of carotenoids, including fucoxanthin, is described briefly and systematically in the new table (table 1) added.
Recently, algae polyphenols have received more attention. Please discuss the accumulation of phlorotanins in algae (e.g. https://doi.org/10.1007/s11094-019-01987-0) and their effect on neurodegenerative diseases.
Answer: This paragraph was added to section 4: “Phlorotannins (brown algae, such as Fucus vesiculosus and Ascophyllum nodosum, polyphenols) are structural analogues of condensed tannins from terrestrial plants and are found exclusively in algae and macrophytes. Phlorotannins are polymers made up of phloroglucinol (1,3,5-trihydroxybenzene) and can make up 25% of the dry algae biomass. They exhibit anti-inflammatory, antioxidant, antibacterial, antiallergic, anti-tumor properties, and inhibit antiplasmin, matrix metalloproteinase, etc. [Obluchinskaya et al. 2019].
A key aspect of the diet is the absorption of its compounds. Algae contain a large amount of carbohydrates. Fucoidan is the main component of brown algae with a broad spectrum of action, the effect of which depends on various factors, including molecular weight (e.g. https://doi.org/10.3390/ijms21239272). Consider the problem of preventing neurodegenerative diseases in terms of the absorption of fucoidan (for example, https://doi.org/10.3390/md16040132).
Answer: This paragraph was added to section 3: Fucoidans extracted from the brown algae Laminaria hyperborea and Saccharina latissima, have recently been suggested as potential therapeutics in age-related macular de-generation (AMD), and in other pathological pathways that include lipid dysregulation, inflammation, oxidative stress and pro-angiogenic signaling [Dörschmann and Klettner 2020]. However, knowing the pharmacokinetic profile of these bioactive molecules, in-cluding their molecular weight, is essential to implement drug development processes [Pozharitskaya et al. 2018].
The use of algal polysaccharides for the treatment of neurodegenerative diseases has been widely considered. Discuss your details with ( e.g. https://doi.org/10.3390/md19020089).
Answer: This paragraph was added to section 3: “Polysaccharide extracts from seaweed have very significant neuroprotective and re-pairing activities. these polysaccharides could be the next great advance in the treatment of neurodegenerative diseases. Fucoidan, ulvan and their derivatives are potential agents to treat Alzheimer's disease, according to a recent review made by Bauer et al. 2021.”
Please discuss the aspects of the use and absorption of other carbohydrates, namely lamiranan, carriginan and others (for example, https://doi.org/10.3390/md18110557).
Answer: This paragraph was added to section 3: “Studies related to the absorption of bioactive compounds extracted from algae, namely polysaccharides, provide vital information on the most appropriate administration routes. If a drug formulation has a high absorption rate, it can be used for the delayed / controlled release of an active ingredient. Understanding the pharmacokinetics of polysaccharides derived from seaweeds (alginates, laminarins, fucoidans, etc.), may lead to their extensive use not only as drugs, but also to improve the bioavailability of certain poorly soluble compounds in pharmaceutical formulations (Shikov et al. 2020).”
For a successful visualization of the manuscript data, provide the data in the form of a table on the use of the algal diet in humans and animals. Compare data from humans and animals.
Answer: We will add a table, organized according the different taxonomic groups of algae (brown, red and green - table 1), where studies with animals and humans, related to the algal extracts and the respective bioactivity and potential treatments for neurodegenerative diseases are summarized.
The manuscript was written by various authors. Please reformat so that it is a single text.
Answer: This review resulted from reading and interpreting several articles and book chapters written by many authors, so the strategy was to compile (without plagiarizing) these texts according to the topics described in the different sections of this review.
Reviewer 3 Report
In this manuscript of "The Seaweed Diet in Prevention and Treatment of the Neurodegenerative and Other Chronic Diseases" authors listed and described the main components of a suitable diet for patients with Parkinson, Alzheimer’s, and Multiple Sclerosis, and other chronic diseases. This review fully demonstrates the authors' accumulation of knowledge in the field and suggests new trends for other scholars to follow. However, I am puzzled by somewhere in the paper. The following are the questions and comments of this manuscript:
- In introduction part, would authors give sufficient reasons for using seaweed as a potential source of neuroprotective agents? Provide enough benefits for these reasons will be better because the importance of the availability and the advantages of seaweeds is not enough highlighted.
- In line 51, please italicize in vitro. Authors need to check for similar formatting irregularities in the remaining full manuscript like using the singular form of “Conclusion” in line 380.
- What is the purpose to show the different species of seaweed in figure 1? In “Seaweeds and Their Neurophysiological Activities” part, can authors try to explain the prevention and treatment mechanism in a specific seaweed species with effects of seaweed activities in a new figure?
- It will be better if authors added a table to conclude various seaweeds species contributing to the pathogenesis for prevention and treatment in neurodegenerative diseases, may include different cytokines or pathways.
- In conclusion, may authors should focus on the prevention and treatment of the neurodegenerative diseases, not only did it appear in the last sentence at the end. It need to prove seaweed diet is useful in neurodegenerative diseases instead of other aspects.
After going through the manuscript, I feel that authors need to revise the paper to reach the level of Marine Drugs.
Author Response
Reviewer 3
In this manuscript of "The Seaweed Diet in Prevention and Treatment of the Neurodegenerative and Other Chronic Diseases" authors listed and described the main components of a suitable diet for patients with Parkinson, Alzheimer’s, and Multiple Sclerosis, and other chronic diseases. This review fully demonstrates the authors' accumulation of knowledge in the field and suggests new trends for other scholars to follow. However, I am puzzled by somewhere in the paper. The following are the questions and comments of this manuscript:
Answer: Thank you very much for the review and comments made. We will proceed with the suggested corrections. All corrections and changes were made in track changes mode
Authors comment: The strategy used to present the information, in the different sections, and since it is very vast, was to present the diverse studies in ascending chronological order. As the method of presenting the bibliography cited in Marine Drugs is by numbers and not by author/year, this strategy is not noticeable!
- In introduction part, would authors give sufficient reasons for using seaweed as a potential source of neuroprotective agents? Provide enough benefits for these reasons will be better because the importance of the availability and the advantages of seaweeds is not enough highlighted.
Answer: This paragraph was added in the end of introduction: “Currently, several lines of study attempt to provide information on the biological activities and neuroprotective effects of seaweed, including antioxidants, anti-neuroinflammatory agents, cholinesterase inhibitory activity and inhibition of neuronal death [5]. Recent studies have shown that microglia activation and the resulting production of pro-inflammatory and neurotoxic factors are sufficient to induce neurodegeneration in animal models. In addition, microglia activation and excessive amounts of proinflammatory mediator release by microglia were observed during the pathogenesis of AD, PD, MS, dementia complex, as well as neuronal post-death in strokes and traumatic brain injuries (Figure 1). Therefore, a mechanism to regulate the release of the inflammatory response by microglia may have important therapeutic potential for the treatment of neuro-degenerative diseases. Several published works point out that seaweed constitutes a relevant source of neuroprotective agents, with particular interest for preventive therapeutics (Table 1) [5].”
- In line 51, please italicize in vitro. Authors need to check for similar formatting irregularities in the remaining full manuscript like using the singular form of “Conclusion” in line 380.
Answer: Done
- What is the purpose to show the different species of seaweed in figure 1? In “Seaweeds and Their Neurophysiological Activities” part, can authors try to explain the prevention and treatment mechanism in a specific seaweed species with effects of seaweed activities in a new figure?
Answer: In order to clarify the mechanisms related to Microglia-mediated neurotoxicity a new figure (1) has been added to the introduction part. The Figure 1, with several examples of species of marine macroalgae used in studies related to the prevention and treatment of neurodegenerative diseases, became Figure 2, also related to the new Table 1 with a summarization of seaweed compounds with bioactivities related to the prevention or treatment of neurodegenerative diseases
- It will be better if authors added a table to conclude various seaweeds species contributing to the pathogenesis for prevention and treatment in neurodegenerative diseases, may include different cytokines or pathways.
Answer: In this sense, Figure 1 and Table 1 have been added, as mentioned above
- In conclusion, may authors should focus on the prevention and treatment of the neurodegenerative diseases, not only did it appear in the last sentence at the end. It need to prove seaweed diet is useful in neurodegenerative diseases instead of other aspects.
Answer: This paragraph was added to the conclusion section: “Several in vitro studies have proven the efficacy of nutraceuticals and food products fortified from bioactive compounds from seaweed, however, the most challenging factor in the food industry is to develop new products that can attract consumers where strange or unfamiliar products are approaching them.”
After going through the manuscript, I feel that authors need to revise the paper to reach the level of Marine Drugs.
Authors: Done
Round 2
Reviewer 1 Report
marinedrugs-1119844
Personally, I think the article is very interesting. I appreciate the effort made by the authors to develop a diet that includes algae products or derivatives of them to prevent diseases of great impact in today's society such as neurodegenerative and chronic diseases.
Some minor comments to previous questions:
In section 3, the scientific names of the species should be in italics. Correct it in all the manuscript. Also, this section contains a lot of information but it appears disorganized. It does not have a common thread. It would also be convenient to create a table with the different studies consulted, the results obtained, the applications, etc.
Answer: The algae scientific names have all been corrected. The strategy used to present the information, and since it is very vast, was to present the different studies in ascending chronological order. As the method of presenting the bibliography cited in Marine Drugs is by numbers and not by author/year, this strategy is not noticeable! We accepted the suggestion to add a table so that the information is more condensed.
The table contains a lot of information, but needs more work. For example, the extraction / compound column is not clear. In some cases, the conditions (e.g., temperature) are indicated and in others not. It would be convenient to make separate columns for extraction method and another for compounds of interest.
On the other hand, I think it is convenient to make individual tables for each large group of algae: green and brown. Especially due to their enormous length (8 pages in the case of brown ones). That is why apart from dividing them into tables according to their color, it is necessary to group the information so that it does not spread out so much. This can be achieved by avoiding duplicates. That is, there are algae that appear referenced in the table several times and sometimes with the same compound of interest. Unify both rows and then put the reference with the corresponding bioactivities in the unified row.
Also, are the studies in vivo or in vitro?
There is a table of the vitamin content of algae, but no information is given on any other compound present in them. Why so important in this table? Why is this done for vitamins and not for example for sugars that have been shown to have antineurodegenerative activity? That table is the object of study and should be made.
Answer: Moreover, we will add an additional table, organized according the different taxonomic groups of algae (brown, red and green - see Table 1)
The reason for such a detailed table of the vitamin content of certain algae remains unexplained when the article focused more on other compounds such as polysaccharides or tannins. What is the content of those other compounds of great interest? Are they really a profitable raw material or are they in such low concentration that their extraction is not viable?
FINAL REMARKS
In my opinion, authors have carried out a really interesting study, with promising expectations for future research. The manuscript is clear and well written. However, I am suggesting MINOR REVISIONS.
Author Response
Reviewer 1 (2nd round)
Personally, I think the article is very interesting. I appreciate the effort made by the authors to develop a diet that includes algae products or derivatives of them to prevent diseases of great impact in today's society such as neurodegenerative and chronic diseases.
Some minor comments to previous questions:
In section 3, the scientific names of the species should be in italics. Correct it in all the manuscript. Also, this section contains a lot of information but it appears disorganized. It does not have a common thread. It would also be convenient to create a table with the different studies consulted, the results obtained, the applications, etc.
Answer: The algae scientific names have all been corrected. The strategy used to present the information, and since it is very vast, was to present the different studies in ascending chronological order. As the method of presenting the bibliography cited in Marine Drugs is by numbers and not by author/year, this strategy is not noticeable! We accepted the suggestion to add a table so that the information is more condensed.
The table contains a lot of information, but needs more work. For example, the extraction / compound column is not clear. In some cases, the conditions (e.g., temperature) are indicated and in others not. It would be convenient to make separate columns for extraction method and another for compounds of interest.
On the other hand, I think it is convenient to make individual tables for each large group of algae: green and brown. Especially due to their enormous length (8 pages in the case of brown ones). That is why apart from dividing them into tables according to their color, it is necessary to group the information so that it does not spread out so much. This can be achieved by avoiding duplicates. That is, there are algae that appear referenced in the table several times and sometimes with the same compound of interest. Unify both rows and then put the reference with the corresponding bioactivities in the unified row.
Also, are the studies in vivo or in vitro?
Answer: Table 1 was completely reformulated in accordance with the reviewer's suggestions, so a new column was added with the extraction methodologies and another column with the tested compounds (when the information is available in the cited article). The works were associated with each of the species used in the studies. When possible, information was added whether studies were performed in vitro or in vivo.
There is a table of the vitamin content of algae, but no information is given on any other compound present in them. Why so important in this table? Why is this done for vitamins and not for example for sugars that have been shown to have antineurodegenerative activity? That table is the object of study and should be made.
Answer: Moreover, we will add an additional table, organized according the different taxonomic groups of algae (brown, red and green - see Table 1)
The reason for such a detailed table of the vitamin content of certain algae remains unexplained when the article focused more on other compounds such as polysaccharides or tannins. What is the content of those other compounds of great interest? Are they really a profitable raw material or are they in such low concentration that their extraction is not viable?
Answer: Table 2 is inserted in the section related to Multiple Sclerosis (MS) and other chronic diseases, as Vitamins are vitally important for these diseases, both in their hypothetical origin and in their treatment, in order to reduce their outbreaks (MS). This importance of vitamins is known by neurologists, and patients with MS, as is the case of the this manuscript first author (see, please, line 378).
FINAL REMARKS
In my opinion, authors have carried out a really interesting study, with promising expectations for future research. The manuscript is clear and well written. However, I am suggesting MINOR REVISIONS.
Answer: The authors are grateful for all the suggestions made by the reviewer, and these were introduced in the manuscript when the original sources (bibliographic references) made it possible to obtain the necessary information.
Reviewer 2 Report
The authors of the manuscript took into account my recommendations and made the necessary changes. I have no more questions.
Author Response
Reviwer 2 (2nd round)
The authors of the manuscript took into account my recommendations and made the necessary changes. I have no more questions.
Authors: Thanks for the comment
Reviewer 3 Report
In my opinion, the revised manuscript can be accepted in the present form.
Author Response
Reviewer 3 (2nd round)
In my opinion, the revised manuscript can be accepted in the present form.
Authors: Thank’s for the comment